# A Look at Culture and Stigma of Suicide: Textual Analysis of Community Theatre Performances

**DOI:** 10.3390/ijerph16030352

**Published:** 2019-01-26

**Authors:** Sarah Keller, Vanessa McNeill, Joy Honea, Lani Paulson Miller

**Affiliations:** 1Department of Communication & Theatre, Montana State University Billings, Billings, MT 59102, USA; 2Department of Psychology, Montana State University Billings, Billings, MT 59102, USA; vanessa.mcneill1@msubillings.edu; 3Department of Social Sciences & Cultural Studies, Montana State University Billings, Billings, MT 59102, USA; jhonea@msubillings.edu; 4Walden University, Minneapolis, MN 55401, USA; lpaulsonmiller@gmail.com

**Keywords:** stigma, suicide, suicidal ideation, cultural competency, health communication, mental illness, depression

## Abstract

Stigma against suicidal ideation and help-seeking is a significant barrier to prevention. Little detail is provided on what types of stigma interfere with help-seeking, how stigma is expressed, and how to reduce it. Five groups of two ethnically diverse community theatre programs were formed to analyze differences in Eastern Montana Caucasian and Native American adolescents and young adults’ experiences with stigma about mental illness and mental health treatment that affect help-seeking for suicidal thoughts and experiences. Over a ten-week period, a grassroots theatre project was used to recruit members from the same population as the audience to write and perform a play on suicide and depression (*n* = 33; 10 males, 23 females; 12 Native American, 21 Caucasian, ages 14–24). Using textual analysis, the community- and campus-based performance scripts were coded for themes related to stigma. Both ethnic groups reported that stigma is a barrier to expressing emotional vulnerability, seeking help, and acknowledging mental illness. We found that Caucasians’ experiences were more individually oriented and Native Americans’ experiences were more collectively oriented. Understanding the cultural bases of experiences with stigma related to mental health treatment for suicide is necessary to create educational programs to reduce stigma for diverse groups of adolescents and young adults.

## 1. Introduction

In 2016, the rate of suicide in Montana was the highest in the United States—almost double the national average at 26 per 100,000 people, compared to 13.4 per 100,000 people nationwide [1]. For Montanans between the ages 25–44, that rate jumps to 40.3 per 100,000. In addition to the “lethal triad” that plagues many rural communities—greater numbers of firearms, drug and alcohol abuse, and limited access to health services [2]—factors exacerbating the distressingly high suicide rate in Montana include long, dark winters, a stoic “cowboy up” mentality [3], lack of mental health awareness [4], and social isolation [5]. This lethal combination not only contributes to suicidal ideation and behavior, but to a precursor factor—stigma [6].

Goffman [7] defined stigma as being discredited by society and condemned to an undesirable social status. Stigma surrounding mental illness, and suicidality in particular, has been documented as an immediate and profound barrier to help-seeking behavior [5]. Research has shown that mental illness stigma reduces patients’ perceived need for help [8], impairs adherence to treatment regimens [9], decreases self-esteem [10], and increases social isolation [11]. 

Stigma can take emotional, cognitive, or behavioral forms. For example, self-stigma is a process in which a person with a mental illness internalizes stigmatizing attitudes and beliefs held by the public [12]. People who see themselves as a burden may believe themselves to be shunned (either by others or themselves), and those who are socially isolated may assume that their condition either results from or contributes to an undesirable social status. Stigma has also been associated with violations of personal freedoms for people trying to access mental health services [13]. Consumers of mental health services have identified forced therapies or treatment, demeaning or infantilizing interactions with staff, dehumanizing procedures in institutions, isolation from community life, and lack of respect for privacy when interviewed about the stigmatizing aspects of mental health programs [13]. By amplifying social differences, stigma exacerbates existing restrictions on individual freedoms typically experienced by individuals assigned a lower social status in society [14]. 

Scholars have identified three primary mechanisms to explain how stigma contributes to suicide. First, according to the stress-coping model of stigma, stigma is seen as a social stressor that promotes “negative emotional reactions, social withdrawal, and hopelessness among people with mental illness, especially if the perceived threat of stigma and social rejection exceeds the coping resources of the individual” [15] (p. 4). Second, stigma contributes to the social isolation of a person experiencing a mental health problem, in part by discouraging interaction and a sense of belonging with others [16]. Social isolation, in turn, is believed to contribute to the risk of suicide, in part, by reducing the desire to discuss one’s mental health status with others [15]. Third, community-wide stigma is associated with individual self-stigma [12]. Studies on predictors of help-seeking have shown that both public stigma and self-stigma are associated with lower willingness to seek help for mental health problems [15] (p. 4). 

Based on comparative studies across countries, Schomerus et al. [15] posited that national variations in suicide rates may reflect variations in cultural beliefs about mental illness and stigma. Furthermore, suicide rates vary by region within national borders, and within ethnic enclaves, reflecting different stressors, levels of stigma, and how normative suicide is seen as a cause of death among community members [15]. Many countries report increased, yet varied, risks of suicidal behaviors among adolescents in immigrant communities [15]. Identified stressors include ruptured family structures, difficulties acculturating to a host country’s cultural and religious traditions, language barriers, and particularly socioeconomic status, which can become a potentially lethal combination [16,17,18,19]. In Montana and elsewhere, Native Americans (NA) have a unique situation in that they experience all these difficulties, yet they live in their country of origin, i.e., they are guests in their own country. Lalonde, discussing Indigenous First Nations Canadians, describes this as “…the special circumstance of finding oneself tugged by both enculturative and acculturative currents” [20] (p. 139). At present, many NA communities in Montana lack sufficient social, emotional, and financial resources, which may increase suicidal behavior risks [21,22]. Nationwide, Indigenous youth are 2.5 times more likely to experience suicidal ideation or attempts than youth from other racial/ethnic youth groups [23,24]. 

Emerging theories of suicide consider how cultural meaning influences suicidality [25]. Culturally, social norms and spiritual beliefs associated with suicidal ideation and attempts influence when and how a person may consider suicide [26]. In anthropological studies, unique cultures apply unique meaning to suicide [25,27]. One culture may believe that suicide is an unforgivable sin, while another culture may view suicide as a socially acceptable solution to grief, pain, or the inevitable end of life [28]. Current theories of suicide predominantly consider “Western individualism”, maintaining that an individual’s motives are factors, but fail to capture non-Western cultural factors [29]. This study differentiates between views of helping others with mental health problems or depression as part of a collective responsibility, versus perspectives that view depression and alienation as a personal problem, where both the experience and the cause are focused on the individual.

### 1.1. Conceptualizing Stigma 

Goffman [7] defined stigma as a process based on the social construction of identity, whereby individuals who are associated with a stigmatized condition are discredited by society and condemned to an undesirable social status. More recently, sociologists like Link and Phelan [30,31] conceptualized stigma as a social process that occurs within the broader sociocultural environment that contributes to structural and institutional discrimination.

Anthropologists have conceptualized stigma as a communal moral judgment, conferring a morally ambivalent status onto the stigmatized group [32]. Kleinman and Hall-Clifford [33] discussed how stigma, for each community, is both locally defined and socially embedded, such that the effects of stigma upon an individual are dependent on beliefs about what constitutes “normal” for that community. Therefore, Kleinman and Hall-Clifford [33] suggested that any intervention to counteract stigma should be rooted in the social network that it targets. “Understanding the unique social and cultural processes that create stigma in the lived worlds of the stigmatized should be the first focus of our efforts to combat stigma” [33] (p. 418).

### 1.2. Approaches to Shed Light on Stigma 

Niederkrotenthaler et al. and Kleinman and Hall-Clifford [5,33,34] called for ethnographic research [30] to shed light on the moral components of stigma, in order to increase help-seeking across heterogeneous populations [5]. Specifically, Walters et al. [35] (p. S105) suggested that interventions be built around risk and protective factors intrinsic to Native American cultures, such as the protective functions of family, community, spirituality, and traditional healing practices [35] (p. S105). Some public health efforts designed to reduce stigma associated with suicide have attempted to raise awareness of the scope of the problem and the prevalence of the threat [36]. However, until we understand more specifically the kinds of stigma that exist within particular communities, and their moral and cultural underpinnings, suicide prevention programs will be limited in their effectiveness in counteracting stigma [37,38,39]. 

Community-based theatre programs can be used to illuminate local cultural attitudes about a health threat [40,41]. As suggested by the Narrative Engagement Framework and the life story elicitation technique, interactive theatre can be used to elicit personal narratives, which can yield important information about perceptions, attitudes, and beliefs within a community. These narratives can also be used to develop messages that are meaningful to other community members [40,41]. This approach has been used successfully in communities around the world to address sensitive health issues [41,42,43]. In step with the Narrative Engagement Framework, actors in this project were instructed to use personal experiences to communicatively engage audience members around the topics of resistance to openly discussing the risk of suicide, the use of professional help, and the strategies needed to intervene in suicidal ideation and attempts.

### 1.3. Cultural Dimensions of Mental Health Stigma 

Cultural differences in the causes and manifestations of stigma, and attitudes towards suicide, are not uniform across populations [44]. Many studies have demonstrated that suicide rates are greater in rural areas than in urban areas [45,46,47,48,49]. When mental illness is combined with low social and/or economic resources, stigma levels (self-stigma and community stigma) contribute to suicidal ideation through factors such as social isolation, hopelessness, and a perception of thwarted belongingness or being a burden [19,22]. Meta-analyses of quantitative and qualitative studies confirm that stigma has a negative effect on help-seeking, thus supporting the idea that stigma-reduction programs require a nuanced approach [8] to meet each unique community’s needs. In the US, many NA communities currently lack social, emotional, financial, and health care resources [20,22]; they may also grapple with cultural prohibitions on the use of professional counseling, which increase stigma and suicidal behavior risks [37,38,39]. NA youth experiencing racial discrimination have higher rates of substance use and suicidal ideation [50]. Yet there is a dearth of literature analyzing NA youth culture and its role in suicide [51,52]. There is a greater prevalence of suicide within NA communities than in other ethnic groups. It is encouraging that a cultural promotion of communal mastery common to most NA cultures (e.g., “I am successful by virtue of my social attachments”) has been shown to increase resiliency in response to stress and reduce depressive mood and anger [53]. While we have not formally investigated this research question, Walters et al. (2002) argued that suicide prevention approaches are primarily Eurocentric in their formulation, disregarding indigenous, communal ways of knowing. Researchers have increasingly [35,54,55] called for incorporating indigenous views into substance-abuse and suicide prevention programs to emphasize cultural strengths such as families and communities, spirituality and traditional healing practices, and group identity attitudes. This article aims to identify culture-specific variables to better inform suicide prevention strategies.

Our research team designed the study presented here to assess young adults’ experiences with and perceptions of stigma surrounding mental illness. We also investigated barriers to accessing prevention services for depression and suicide. The study involved a textual analysis of the scripts from community- and campus-based theatre performances written and produced by students in Eastern Montana high schools and one state university in 2012–2015. The objective of this process was to shed light on attitudes towards suicide prevention resources among members of the participating community—both the writer-actors and the audience members. We analyzed transcripts of the performances to gain a systematic understanding of the messages they contained. This paper addresses two research questions (RQs 1 and 2) that were developed prior to the project, and one research question (RQ3) that emerged after the plays were performed:RQ1. What mental health issues are reported by Eastern Montana adolescents and young adults?RQ2. How does stigma about mental illness and mental health treatment affect help-seeking behavior among Eastern Montana adolescents and young adults?RQ3. What are the differences in collective vs. individual orientation expressed by NA and Caucasian adolescents and young adults?

## 2. Materials and Methods

### 2.1. Theatre Project 

The performances under analysis in this study originated through five theatre workshops to address the risk of suicide. The theatre workshops were conducted by professional theater directors (with Master’s of fine arts degrees) recruited from the university’s theatre department, in conjunction with licensed professional counselors at each workshop location. The theatre directors worked with a project coordinator to recruit student volunteers for the plays at high school and college venues. Volunteer writer-actor groups met for 2–3 h biweekly at prearranged after-school locations or university campus facilities for 10 weeks to share personal experiences with suicide and/or major depression; each group collaboratively wrote a unique play based on their members’ experiences. Throughout the playwriting process, student-actors shared memories, songs, and poetry to develop a creative script based on their own experiences with depression, suicidal ideation, suicide attempts, and grief over a friend’s or family member’s suicide. The narrative development process included any part of the students’ life experiences related to suicide and depression (including among friends and family) they felt comfortable sharing. At the end of the writing–rehearsal period, the students staged performances for their peers to demonstrate how communities could talk among themselves and with mental health professionals about mental illness, suicidal ideation, loss, and prevention. Montana State University Billings (IRB00001622) approved this study.

Community-based programs that engage community members in storytelling and performance, such as this theatre production, offer a potentially effective approach to revealing underlying attitudes and beliefs about sensitive issues, and can be used as a research tool to generate interpersonal dialogue [40,41,43]. Our team believed that a play produced, written, and performed by individuals with intimate, personal knowledge about mental illness and suicide would illuminate community-specific factors associated with stigmatization of suicidal conditions. We used the theatre performance as a method for identifying underlying barriers to and solutions for reducing the state’s suicide crisis [40,41,42,43]. 

### 2.2. Subjects 

At each participating high school and university, subjects (*n* = 33) were recruited via classroom announcements and word of mouth (face-to-face recruitment), asking volunteers to take part in writing and performing an original play about suicide and depression. Two of the five theatre workshops were recruited from Native American high-school after-school programs (HS 2 and HS 4); the remaining three theatre workshops comprised Caucasian adolescents and young adults from local high schools and one college (see Table 1 for demographic information). Potential participants were informed that the plays would be autobiographical in nature and video-recorded for a professional documentary. Subjects were informed of their right to withdraw from the study and were given both participant and parental consent forms to read and sign before starting the 10-week rehearsal and production process. About one-third of initial participants withdrew from the project prior to its completion citing logistical and time constraints, leaving 33 subjects remaining across the five workshops. Recruitment relied on volunteer efforts without compensation on the part of participants. Unlike traditional focus groups, the theatre research method used here asked for 2–3 h per week from participants over a 10-week period. The length of commitment requested likely contributed to high attrition. The remaining sample was comprised of 10 males, 23 females; 12 of whom were Native American, 21 were Caucasian; 28 were high school students, 5 were college students; and 5 of whom came from a rural frontier community, while the rest were urban (Table 1).

### 2.3. Narrative Development 

Workshop meetings were held at high school and college campus locations (e.g., auditoriums, empty classrooms). In addition to the theatre directors and counselors, researchers were present once per week to introduce themselves and observe the script development process. Field notes were taken during these observations, but no audio or video recordings were taken until the final performance of each workshop. The researchers’ presence was restricted to ensure that students had an opportunity to develop narratives without the potential inhibition caused by a researcher’s gaze. For the first meeting, an outside specialist administered QPR (question, persuade, refer) suicide prevention gatekeeper training, which provided the initial platform to talk about such a personal and painful topic. At the rehearsals, theatre directors gently probed the writer-actors’ experiences with stress, anxiety, depression, alienation from school and home, and relationships. These prompts were not standardized across the directors; each director was given the freedom to develop their own prompts and questions to engage participants in discussion around the themes of suicide, depression, loss, and help-seeking. As related themes emerged, the directors were encouraged to incorporate these subthemes into the emerging scripts. By contributing creative non-fiction based on their own experiences—with self-harm (e.g., cutting), suicidal ideation, suicide attempts, and grief over a friend’s or family member’s suicide—the writer-actors collaboratively created an original narrative theatrical performance [41,56] (p. 658).

### 2.4. Textual Analysis 

Once the scripts were complete, a team of researchers (study authors) (three with PhDs and one with a Master’s degree in social sciences, all female) was convened to analyze the text. All members of the team had extensive experience with qualitative research methods and analysis. The research team members all had a part in designing and implementing the community-based theatre project. Student writer-actors recruited for the plays were introduced to the research team members and their role was explained. 

The team conducted a textual analysis, using grounded theory and content analysis, because established qualitative data analysis techniques [56] would allow us, as researchers, to draw inferences from the rich datasets that the scripts of the students’ plays comprised [56,57,58]. Textual analysis provides a means for examining the values and norms that underlie the message content [59]. This process involves describing the content, structure, and functions of the messages contained in texts.

Between 2012 and 2015, the research team recorded video footage from the five theatre performances described above, along with conversations among performers and audience members in post-play question and answer sessions. The textual sample consisted of the entire population of such plays produced by the project under study. Digital video recordings and transcriptions of each play were entered into the qualitative analysis software MaxQDA (VERBI GmbH, Version 11, 2015). None of the transcripts were returned to the theatre workshop participants for feedback. 

#### 2.4.1. Code Development

Three members of the research team (one professor and two research assistants trained in qualitative research) made up the coding team. One of the five play transcripts was selected at random to serve as a subsample for initial code development. The team’s analytical approach was informed by coding methods laid out by Miles and Huberman [60], Corbin and Strauss [61], Fairclough [62], and DeCuir-Gunby, Marshall, and McCulloch [63]. The team began developing codes by listing general categories of conceptual variables. These a priori categories were constructed during an earlier focus group study examining the impact of the plays on audience members and writer-actors, addressing research questions pertaining to whether the plays increased perceived self-efficacy to seek help [22]. While some participants were the same as those used in this study, the current study focused exclusively on comments articulated through the playwriting process, while the focus group study analyzed comments collected after the plays were completed. The coding team clarified the operational definitions of the a priori categories and collectively reviewed several examples of each to ensure uniform interpretation of the definitions. The complete list of a priori categories is included in Appendix A.

After achieving consensus on the main categories, the team began open-coding a subset of the transcripts, using both the a priori categories, and identifying additional ideas as they emerged. As needed, the team refined the categories, re-naming codes [62] and using axial coding [61], to identify how passages in the data might be arranged into main- and subcodes (Table 2). As the team worked through the coding process, they noted that the voices in the transcripts varied substantially in whether they identified problems and solutions as individual or communal in orientation. As such, the team adopted RQ3 to compare communal vs. individual orientations, and created a supra-code across all categories, so that they could classify comments accordingly. This iterative cycle continued through several meetings until the team was satisfied that they had achieved data saturation (such that all concepts identified in the text were captured by coding categories) and consensus on coding definitions.

#### 2.4.2. Coding

The team first practiced coding using subsamples of the video footage to become familiar with the variables and to resolve emergent questions. Once the codebook was established, the coders worked independently to assign codes to all the remaining performance transcripts. Codable passages (*n* = 452) included any scenes or statements in the transcripts that fit into one or more of the main categories, according to the coders. Each coder identified the main category/-ies and subcategory/-ies that best fit each passage. 

Reliability was assessed by measuring the agreement between each of the independent coders. Based on Wimmer and Dominick’s [64] recommendation of using a sample of between 10% and 25% of the study text in order to perform reliability testing, coder and intercoder agreement was measured using Krippendorff’s alpha (α) [65] reliability estimate for judgments. Initial reliability scores were low, leading to in-depth discussions among the entire research team and referencing the codebook when necessary, to discuss themes and their interrelationships [66]. After these discussions and recoding when necessary, the research team calculated interrater reliability at 83% (Table 3).

## 3. Results

The coding team identified and coded a total of 452 passages of text from five (5) performances (Table 3). Passages were frequently coded in more than one category. The most common main category identified (which includes subcategories laid out in Table 1) were “self-reports of mental health issues” (*n* = 153); “communal vs. individual orientation” (*n* = 52); and “stigma” (*n* = 94). Stigma was measured both as a standalone main category and as a subcategory under “barriers to help-seeking” and “factors of suicide” (Appendix A). Additionally, subcategories conceptually related to the construct of stigma emerged throughout the codebook, e.g., hopelessness under “self-reports of mental health issues”. 

### 3.1. RQ1: Mental Health Issues

In response to RQ1, “What mental health issues are reported by Eastern Montana adolescents and young adults?”, the most frequently identified main coding category from the transcripts consisted of self-reports of mental health issues (*n* = 153, 34%, see Table 4). Subcategories of mental health issues were identified in the coding and disaggregated to gain a richer understanding of this phenomenon. The majority of experiences reported were feelings of hopelessness (*n* = 32), that is, a person’s belief that they have no options to improve their own situation. Hopelessness is illustrated in the statement, “It’s not fair… there’s, there’s no place to go. Nowhere to run, no one to turn to. I’m completely lost” (Caucasian, college, female). 

The prevalence of experience with hopelessness reported by students in the community plays led us to consider why students feel hopeless, and what relationship that might have to suicide, or stigma related to mental health issues. An individual who feels their mental health status is stigmatized by others will be more likely to feel they have no one to turn to. Given previous findings relating stigma to social withdrawal, limited interaction with others, and self-stigma, people who feel stigmatized might also feel hopeless due to their perceived inability to reach out for help [12,15,25].

Depression (*n* = 19) was the second most frequently coded subcategory, exemplified by the statements:
My brother [died by] suicide a few years ago. I fell into depression. Today I still have depression, but it’s not as bad as it was. I still miss my bro, always….(Caucasian, college, female)
How do I get self-esteem up to par where I can feel like I’m worth even in my adult age? How do feel like I’m worth what other people see in me?(Caucasian, high school, female)

Above, we hear two distinct articulations of depression and experiences of low self-esteem. In the first quote, although not suicidal, deep grief leads to depression that is serious enough for the young woman to describe on stage. In the second quote, a young woman struggles with finding a sense of self-worth that meets the praise and expectations she has received from others. 

Depression related to grief was described by Native American participants, as well as a view that depression is a common experience in adolescence:
What the parents need to know is that kids do get depressed. It’s a natural way of life—you get depressed, you sit there and eat your carbs or whatever. I did that. That was bad. So I did that and I finally got out of my hole the last [month].(Native American, high school, female).

Suicidal ideation was the third most frequently mentioned mental health issue (*n* = 18), exemplified in passages such as:
I wish I could get this all done and over with. It’s just so hard living in hell. Maybe I should end it or maybe not. What should I do?(Caucasian, college, female)
I’ve been in dark places myself and I’ve thought about it. And, and it was actually the concern over my child that kept me from doing anything. And I didn’t probably, I do understand how despair can be, you know, I can understand how she could feel alone. But she, she shot herself in the face.(Caucasian, high school, female)
As I’m driving, I’d like to just drive off the road. Or when I’m in the kitchen, if I could just use a knife or when I’m home alone. If I could just get to my dad’s gun(Caucasian, high school, male)
Why do I think of death when I should think of life? Why do I feel sad all day?(Caucasian, high school, male)
I was literally worried about my friend, she was literally about ready to commit suicide that week and she told me that she was going to. I literally went to my counselor and I’m like, “I don’t know what to do. This weight has been on me like for a week now.”(Native American, high school, female).

In all of the above quotes, young people talk about their lives as consisting of “hell”, “dark places”, and “think[ing] of death”. Perhaps more alarming, each expression of serious discontent is accompanied by explicit talk of taking one’s own life, or of a friend committing suicide. While it is quite possible such experiences are inherent to adolescence and do not necessarily indicate a risk of suicide, the correlation of these comments with epidemic rates of adolescent suicide is concerning.

### 3.2. RQ2: Stigma

Although constructs related to stigma were coded as subcategories under several main code categories, segments were also coded as reports of stigma as a standalone category. Segments were selected for this category if they exhibited reports of: (a) Public stigma (exogenous origins, e.g., stereotypes, prejudices, social derogation exhibited in home, school, church, and other public settings); (b) internal stigma, known as self-stigma (endogenous beliefs), including fear of being shunned, experience of being shunned, disconnection from others, thwarted belongingness, perceived burdensomeness; and (c) stigma of help-seeking, which was evidenced through comments that we labeled “defensive-avoidance” [67]. 

A total of 94 (21%) of all segments were coded under the main category of stigma, pertaining to reports of what is being stigmatized, as opposed to reports of experiences with stigma. Emotional vulnerability, needing help, mental illness, and mental health treatment emerged as subcategories of stigma. 

In response to RQ2, “How does stigma about mental illness and mental health treatment affect help-seeking behavior among Eastern Montana young adults?”, our results indicated that defensive-avoidance reactions fueled fears that others are negatively judging emotional vulnerability and mental health treatment and may constitute a significant barrier to help-seeking among Eastern Montana adolescents and young adults. The literature has established stigma as a well-documented factor contributing to suicide [4,15]. Based on their lived experiences, the writer-actors in this project seemed to know this to be true.
Suicide, a touchy subject no one wants to talk about. But it needs to be talked about. It seems no one wants to talk about it until it’s too late. Suicide affects everyone, teachers, families, friends, and even co-workers. But the teens are affected the most. Still, no one wants to talk about it.(Caucasian, high school, female).

By contrast, other barriers to help-seeking, such as limited access to resources, and low awareness of how to find help, were not frequently found in the transcripts.

#### 3.2.1. Public Stigma

Of the passages coded as stigma, 23 (5%) were coded as public stigma. Below, one female high school student revealed that her sense of alienation is based on her perception of being rejected by others. The quote represents the cyclical relationship between perceived public stigma (against mental health problems) and self-proscribed deviant behaviors or attitudes that contribute to self-stigma.
My collection is different. Some would say it’s odd, and most heads turn away when they see, but I don’t care. Well, I collect scars. Crazy, right? It’s true though. Scars in all shapes and sizes. You see, I don’t like collecting that happy crap; I collect what matches me: depressed, angry, completely helpless. I collect scars because it’s who I am. Of course I notice reactions. All eyes are on me, or some just refuse eye contact period. But it doesn’t make a difference. They both mean the same thing. My collection is frowned upon in society.(Caucasian, high school, female)

Another comment reflected students’ perceptions that other people are negatively judging them: “Look at all those people out there, staring at you, waiting for you to mess up like you always do” (Caucasian, college, female). Clearly, the interplay between external judgment and internal fears is a blurry one. When external judgement involves societal discrimination, the line between internal and external judgments becomes clearer. Native American participants described racism as an added layer to the stigma they experience as individuals. One NA audience member responded to the play by commenting:
We saw a lot of these teenagers depicting very seemingly and realistic scenes of dealing with the stresses that teenagers have in their daily life, and also dealing with many of the roles and stereotypes that they perceive are forced upon them. Now you take all of that and add an additional layer of the stereotyping that is felt by the teenagers in our Native communities.(NA high school male, audience member)

#### 3.2.2. Self-Stigma

Of the passages coded as stigma, 46 (10%) consisted of expressions of self-stigma, which we subcategorized into: (1) Fears of being shunned; (2) thwarted belongingness; and (3) perceived burdensomeness. 

*Fears of being shunned.* Comments reflected fears of being shunned for accessing treatment, fears of being shunned in general, and experiences with being shunned. For example, one segment coded both as “fears of being shunned” and “self-stigma” consisted of:
How come they think I’m full and funny, but I don’t? Why do I wear a mask over my emotions? Why do I seem happy but inside am dark, too dark to see the faintest light? Should I end it all right now?(Caucasian, high school, male)
Here, a high school student explains how he does not feel like he fits in with his peers, and, specifically, that he has to hide his darker or negative emotions in order to be accepted. These thoughts lead directly to thoughts about suicide. An example illustrating the perception of stigma against admitting one’s emotional vulnerability appears in this passage: “Should I let them [emotions/thoughts] out? What if people laugh? What if people cry?”(Caucasian, college, female)

*Thwarted belongingness.* Two coding categories that were categorized as self-stigma are “thwarted belongingness” (*n* = 31, 7%) and “perceived burdensomeness” [68]. Thwarted belonging occurs when one would prefer to feel like a valued member of a community but is socially isolated, and the basic need for human connectedness is unmet [69]. The following passages illustrate explicit feelings of thwarted belongingness:
Nobody cares about me, not a damn person. My teachers all hate me because I’m not good enough for them and, and, and… I’m only useful to my dad because he can yell at me. No one listens to me. I’m completely alone. And when I’m all alone, all I have for company is myself. And lately I haven’t been very great company.(Caucasian, high school, male)

In the passage above, a young man discusses his feelings of not belonging in his immediate circles of social support, due to verbal aggression from his father, combined with a sense of underperforming at school and a perception that he does not have any close friends.
Yeah, I know what it is. ”That kid is a jerk.” I’m a jerk. I’m a bully, a meanie just for picking on that kid. Trust me, what I do to that kid or whatever is nothing compared to what I get at home. My parents fight, then they put it on me. Then my mom beats me after my dad leaves… not just physically but mentally.(Caucasian, high school, female)

Above, a young woman talks about not fitting in because she reports that she is replicating abuse she experiences at home. In this quote, we can see the direct connection between family dysfunction or violence and the experience of thwarted belongingness.

*Perceived burdensomeness.* The corresponding construct to thwarted belongingness is perceived burdensomeness. Perceived burdensomeness is a mental state characterized by perceptions that others would “be better off if I were gone”, which manifests when the need for social competence is unmet [70] (p. 197). The following passages illustrate examples:
Now the worst thing is, while I love my parents, and how are they going to deal with this [my suicide]. You know, maybe I shouldn’t, no. No, no, no, things’ll be better this way. Soon they won’t have to deal with their nothing of a son and, and nothing will leave this world, just, just as nothing entered it. People will go on with their lives, and lives and, and maybe, maybe I shouldn’t…(Caucasian, high school, male)
[My mother] gets inside my head, makes me feel like I’m nothing, makes me feel like I should just jump off a bridge(Caucasian, high school, male)

In these two examples, individuals express their sense of being a burden to both their immediate social circles and society at large. The overlap in descriptions of thwarted belongingness and perceived burdensomeness is a direct result of the intricate relationship between these two feelings as experienced and described by adolescents.

#### 3.2.3. Stigma of Help-Seeking 

Among the passages coded as stigma, nearly half were categorized as stigma of help-seeking, and most of these were defined as “defensive-avoidance” (*n* = 25, 6%). “People try to control their fear by suppressing thoughts of the danger (defensive-avoidance) or by reacting against the communicator or message (i.e., perceived manipulation, message minimization)” [71] (p. 116).

For example, the comment below depicts the stigma surrounding mental health and the challenges of trying to reach out to encourage someone else—in this case, a fellow Native American—to discuss emotional vulnerabilities:
Whether or not you [sic], on the reservation, [sic]… ask [people] like “what’s your problem?” … they’ll have trouble with, you know, reading your song, and then like, you know, they get all defensive.(NA, high school, male)

Despite what the literature shows about collective agency among Native Americans, the above quote illustrates that fear of disclosure is prevalent in both tribal and nontribal communities, and that this fear is sometimes masked by defensive or angry responses. Similarly, a Native American female adolescent described her hesitation to seek professional help on behalf of a friend who was suicidal:
And [the counselor’s] like, “Why didn’t you come to me sooner?” and I’m like “I had to think it over in my head, what I was gonna say and if I should come to you or help her not to, like, do anything.”(Native American, high school, female)

We subdivided the defensive-avoidance category based on expressions of defensiveness, denial, or reactance (blaming the messenger, e.g., “I’m not the one with the problem, I think you have a problem”), collectively made up 27% (*n* = 25) of the 94 stigma codes. Passages coded as behaviors symptomatic of defensive-avoidance reactions included truancy (skipping classes), alcohol and drug abuse, and cutting (self-harm). For example, the following segment of dialogue among three high school students was coded as truancy and alcohol abuse—both defensive-avoidant behaviors:
Female 1: “So I heard this girl call saying you’ve been skipping classes.”
Male: “No, I haven’t been doing that. I’ve been going to all my classes.”
Female 2: “Sure, and you haven’t been drinking your dad’s beer.”

Another segment from a college performance offers deeper insight into young adults’ substance abuse: “You know, the great thing about drugs is that when you’re doing them, nobody asks you why. Nobody wants to know how you got started” (Caucasian, college, female). 

Witte [67] asserted that individuals will pursue negative or harmful coping strategies when their perceived threat overwhelms their perceived efficacy to achieve a solution to a problem. In this case, the pursuit of drinking, truancy, substance abuse, etc. at the very least are unhealthy coping methods for experiences of discontent. More significantly, they are actions that interfere with constructive help-seeking for that discontent, namely the pursuit of professional counseling, disclosure to supportive others, engagement in healthy activities, and/or other types of mental health treatment.

### 3.3. RQ3: Communal vs. Individual Orientation

In response to RQ3, “What are the differences in collective vs. individual orientation expressed by NA and Caucasian adolescents and young adults?” a total of 52 of the 452 passages were coded as having a strong communal or individual orientation. Communal orientations were ascribed to passages designating both the locus of a problem and the solution in the community, rather than the individual. Individual orientations were defined as passages indicating the perceived problem as located within an individual or pertaining specifically to individual characteristics. “Individualism implies that the self is permanent, separate from context, trait-like, and a causal nexus; that reasoning is a tool to separate out main points from irrelevant background or context; and that relationships and group memberships are impermanent and non-intensive” [72] (p. 12). Conversely, collectivism implies that the self is “malleable and context dependent” and that relationships and group memberships are “ascribed and fixed” [72] (p. 12).

A strong ethnic difference was revealed in the collectivist vs. individualist passages. An example of collective notions of depression and mental health problems was articulated by one Native American male:
Okay. Well, Natives, we are different. We don’t really open up as much as we should. And a part of that is because growing up on the rez, you’re, it’s, it’s hard. It may seem difficult and, so until you’ve actually lived there. You know that it takes skills to survive on the rez. When you’re brought up with sort of this—defense mechanism, an inner, inner sort of system of defense they use.(Native American, high school, male)

In the above quote, a young man talks about how members of his tribe share a common tendency to hide their emotional vulnerabilities. What is interesting is that he depicts this challenge (fear of disclosure) as a common problem, not an isolated affliction. By contrast, Caucasian students in our sample described their experiences with depression as a personal problem, one that is partly caused by the rejection or negative judgment of their immediate circles of support. Not only do we not hear Caucasian students describe a maxim requiring them to help each other out, but just the opposite, they speak frequently of rejecting others and being rejected by them. An example was given by a Caucasian high school female:
I know exactly what you all think I am. I’m some Judd Nelson character with my fists in the air and I smoke to rebel from the social norm. I don’t give a damn about the social norm…. I mean, I really don’t care what they think of me so they can pinch their noses or say their prayers or whatever, but I’m going to keep doing whatever the hell makes me feel not like crap, and they can come to whatever conclusions they want. But that they think for a minute that they know who I am? They’re dead wrong.(Caucasian, high school, female)

Caucasian students described their experiences with mental health crises from an individual perspective—they feel that they must suffer alone and help themselves recover from the crisis alone. This orientation is prevalent, even though the writer-actors know that seeking help from others is imperative for recovery.
I would say it’s the words that hurt most, but honestly it’s the silent judgment that comes from the peering eyes of those around me. It’s the whispered words that appear as my back turns to the cowardly. It’s the complete lack of understanding that takes place in this melting pot of awful. Now we’re asked to be who we are and embrace the weird, but how can we do that when society demands we reject ourselves?(Caucasian, high school, male)

In identifying solutions to the suicide epidemic, Native American students talked about group or public efforts, such as ceremonies—including the current theatre performance—as pathways to change. The following quote reflects a group and familial identity in relation to suicide:
I think it’s part of what we are longing here is what we’re doing this evening. I then have to share with you my own experience with depression and suicide attempts in my family in order to serve as your master of ceremonies.(Native American, high school, female)

By contrast, Caucasians tended to describe the potential solution for alienation as rooted in another individual, rather than a societal intervention:
We reject the weird and soon the unreasonable hate of others becomes the unreasonable hate of ourselves and then what? Do we just fly away like Superman? No. No, there is no Superman. No one to shield us from those words that sting like bullets. No one to save us, the weird, the outcasts, those just trying to hang on.(Caucasian, high-school-aged male)

Another strategy for individualized suicide prevention was articulated by a Caucasian female high school student who described the solution to her depression as coming from friends who care and parents who leave her alone:
I want a different path. One where perfection is easy. One, one where my parents don’t tell me I’m going to hell whenever, whenever I’m not perfect. When we have real friends, ones who care, ones who don’t tell me that I don’t have the guts whenever I confide in them.(Caucasian, high school, female)

Meanwhile, NA students expressed an intergenerational and spiritual orientation towards suicide prevention, reflecting the group identity continually found in our sample:
My grandpa Peter once said that in order to get your way in this life, you know, you have to help each other out. And later on in life, the creator will help you on the days that you will be sad. And the days that you will become ill or just physically and spiritually sad…and that we all gotta help each other to get into this world and, you know, help each other out.(NA, high school, male)

In this statement, we hear family dynamics (“grandpa”) and larger spiritual community (“the creator”) both calling upon people to “help each other out” on both ends of life, birth, and death. According to this young man, the NA community (both in a familial and spiritual manner) moral code calls for communal helping of others.

Based on the above, in response to RQ3, “What are the differences in collective vs. individual orientation expressed by NA and Caucasian adolescents and young adults?”, we found that the cultural differences in how Native American and Caucasian young adults view stigma is rooted in their perspective of mental health problems, and depression in particular. Native American youth tended to speak of help as something that is the responsibility of each member of their community, and the barrier was described as the failure of community members to step up to the plate and help each other. By contrast, Caucasian students described mental health problems as individual in nature, and the barriers to help as caused by specific individuals’ (e.g., parents, teachers, friends) failure to care.

Similarly, Caucasian youth tended to describe depression and alienation as an individually unique experience, typically caused by one person not fitting into the social norm or being rejected by significant others in their life. By contrast, Native American students described depression as something their entire tribe (or even everyone) experiences, and that the feelings are common.

Finally, the collective vs. individual orientation represented a difference in how each group viewed suicide. We found that Native Americans were more likely to see suicide as a result of not living in the present, failing to let go of painful experiences, and failure of others in the community to reach out and help one another. Caucasian students described the cause of suicide as a result of being rejected by others, not belonging, not fitting in. 

## 4. Discussion

An empirically based, thematic textual analysis verified the important role that stigma—both public stigma and self-stigma—plays in preventing young people from seeking professional help to reduce the risk of suicide [4,8]. 

In response to RQ1, “What mental health issues are reported by Eastern Montana adolescents and young adults?”, salient mental health reports included hopelessness and depression. Together with the stigma described in the Results section, these psychological states reveal a web of challenges that permeates many aspects of adolescent and young adult lives in Eastern Montana in relation to mental health and help-seeking. Our study also showed that hopelessness, the belief that one is unable to do anything to resolve a health threat (similar to defensive-avoidance), was the most common self-reported mental health problem by both Caucasian and NA youth. Here, it is possible that the self-reports of mental health challenges may reflect our findings on stigma—an internal and external pressure to avoid violating local cultural norms—that may prevent young people from varied ethnic backgrounds from seeking treatment for their mental health challenges and increase the likelihood of hopelessness. 

In response to RQ2, “How does stigma about mental illness and mental health treatment affect help-seeking behavior among Eastern Montana adolescents and young adults?”, the analysis revealed prominent barriers to help-seeking involved both public and self-stigma (defensive-avoidance and thwarted belongingness). Many of the experiences reported by adolescents and young adults in this study were common across ethnic and racial lines. Fear of being publicly stigmatized seemed to clearly limit a person’s willingness or ability to seek help for a mental health risk or experience of suicidal thoughts. 

Self-stigma was also shown to mitigate a person’s willingness to take any action on their own behalf when confronted with a health threat—especially when they perceived the threat to be stigmatized. Corrigan, Larson, and Rusch posited that self-stigma results from internalized stereotypes and prejudices, which in turn lower self-esteem and confidence, resulting in reduced achievement and abandonment of goals, i.e., the “why try” effect, ultimately ending in lowered self-efficacy [73]. Thus, it is possible to construe defensive-avoidance, a fear control response that occurs when an individual feels unable to control a risk [65], as both a result of self-stigma and a barrier to help-seeking. We believe that the perception of public stigma against mental illness and mental health treatment—identified in our coding for stigma—may have contributed to these defensive-avoidance reactions.

In examining self-stigma in this analysis of the youth theatre performances scripts, the construct of thwarted belongingness did reveal differences along ethnic lines. The feeling that one does not belong among one’s peers or significant others and the experience of social isolation was frequently mentioned or described by Caucasian youth but not Native Americans [69]. While Native Americans continually described the need to help each other, Caucasians talked about both rejecting—and being rejected by—others. In explicating this construct of thwarted belongingness, our notions of stigma again play a role. How and why a person comes to feel socially isolated seems to be intimately intertwined with the construct of self-stigma, whereby a person internalizes perceptions of public stigma to the extent that they no longer feel capable or worthy of interacting with society [70].

In response to RQ3, “What are the differences in collective vs. individual orientation expressed by NA and Caucasian adolescents and young adults?”, our results indicated that stigma, suicidal ideation, and help-seeking are described quite differently by Caucasian youth compared to NA youth. While NA participants described experiences of loss, suicidal thoughts, and helping others as both an affliction and an obligation of each member of a community, Caucasian young people described their experiences with suicidal thoughts as an individual affliction typically caused by alienation from their immediate circles of support. Although it is a mistake to assume Indigenous people fall under one category or one culture, and although each NA nation has its own unique cultural beliefs and traditions [74], one key factor that is a core value across NA tribal nations is the expectation that “Native people help each other. This norm of mutual support is an important part of Native identity and traditional culture” [75] (p. 143). “Mutual support not only is the preferred method of assistance for youth struggling with suicide, but also is an integral part of American Indian identity and culture” [75] (p. 145). This communal orientation may have implications for the development of culturally competent suicide prevention programs.

However, it should be noted that while we did not see direct reports of racial discrimination, Native Americans experience suicide at 1.5 times the national average and are ethnic minorities in the United States [1]. As such, researchers and practitioners need to pay close attention to how their experiences of stigma, suicidal loss, hopelessness, and help-seeking are impacted by their minority status.

A limitation of our research stems from the qualitative nature of our approach. Because the majority of textual analysis involves the subjective interpretation of data, care was taken to ensure the validity and reliability of results. Construct validity was achieved through clearly defined research questions and operationalized themes and concepts, while reliability was ensured via the use of multiple coders. Data generated via textual analysis cannot be generalized to an entire population and, because it relies on subjective human beings as the research instrument, it cannot be assumed to be completely objective; nevertheless, there are many strengths of textual analysis as a research method. It generates rich, detailed data not always possible with more common methods like surveys. In addition, much of the information used in textual analysis can be gathered unobtrusively, so that the act of being studied does not influence the behavior of the subjects.

A second limitation of our work is the use of a superficial dichotomy between individualist and collective cultures. While it is indeed true that cultures vary widely in their orientation regarding collectivist vs. individualist thinking, it is impossible to draw far-reaching conclusions about an entire culture based on the few comments analyzed in this text. Furthermore, subcultures vary widely within parent cultures; hence, we are not able to draw conclusions beyond the small group of students interviewed, much less their entire tribes or communities, and we certainly cannot extend or apply our conclusions to entire races or ethnicities. Nonetheless, the contrasting orientations under discussion may have implications for individuals’ ability or inclination to help others at risk of suicide [72].

## 5. Conclusions

We feel our work adds to a body of research on interpersonal storytelling, narrative medicine, and suicide prevention research. Our project is novel in that it qualitatively categorized voices articulated through a community-based suicide prevention setting. The hope is that community theatre of this type can provide a research tool for identifying life stressors and sources of cultural support that may ultimately contribute to programs to increase the use of suicide prevention resources in remote communities where counseling and discussion of mental health problems are largely taboo. By identifying the particular barriers that young adults of varying ethnic backgrounds face in accessing mental health treatment and interpersonal communication about suicide and depression, we hope this paper may provide a crucial link between exhibited warning signs and access to care. 

Our findings supported prior research indicating that communal mastery is common among NA cultures. We feel, based on these findings, that such a collective orientation should be emphasized as a cultural strength when designing suicide prevention programs targeted at NA communities. As mentioned, communal mastery has been shown to increase resiliency in response to stress and reduce depressive mood and anger [51]. If it is indeed true that most suicide prevention approaches are primarily Eurocentric in their formulation, more attention needs to be paid to indigenous, communal ways of knowing. We echo previous researchers’ [49] calls for incorporating indigenous views into suicide prevention programs to emphasize cultural differences in communication styles, identity development, perceived locus of control, and attribution [76].

These findings add a new dimension to existing research on stigma against mental illness, which, to date, has mostly addressed the development of scales to measure stigma, models to increase our understanding of how to diminish stigma’s impact, and explanations of stigma by examining the social cognitive elements of the stigmatizer [77]. By explicating the construct of stigma in its relationship to help-seeking behavior and suicide, we hope to contribute to future research on the precise expressions and experiences of stigma among communities and the development of more effective interventions to counteract it. We recommend that approaches similar to the community-based theatre program described here be used to identify community-specific expressions and experiences with stigma in relation to mental health in order to develop tailored interventions for suicide prevention. 

## Figures and Tables

**Table 1 ijerph-16-00352-t001:** Subjects by location, gender, and ethnicity.

Workshop	Male	Female	Native American	Caucasian
HS 1	1	5	0	6
HS 2	3	5	6	2
HS 3	3	5	0	8
HS 4	3	3	6	0
College	0	5	0	5
Total	10	23	12	21

**Table 2 ijerph-16-00352-t002:** Main coding categories, subcategories, and descriptive themes.

Supra-Code	Main Codes	Subcategories	Examples
Communal vs. individual orientation	Stigma	Public stigma Self-stigma Stigma of help-seeking	Thwarted belongingness Stigma about mental illness Emotional vulnerability Experience with stigma
	Self-reports of mental health issues	Hopelessness	
		Depression	
		Suicidal ideation	
		Self-harming	
		Isolation (loneliness)	
		Uncontrollable thoughts	
		Helplessness	

**Table 3 ijerph-16-00352-t003:** Percentage agreement between coders and percentage of passages in which coding categories appeared (See Appendix A for all coding categories).

Category		Percent Agreement	Frequency in Transcripts *n* (%)	
Stigma	Public stigma	82.1	23 (5)	94 (21)
	Self-stigma	80.2	46 (10)	
	Stigma of help-seeking	88	23 (5)	
Self-reports of mental health		83.1	153 (34)	
Communal vs. individual orientation		88	52 (12)	

**Table 4 ijerph-16-00352-t004:** Self-Reports of mental health issues.

Self-Reports of Mental Health Issues	Frequency in Category *n* (%)
Hopelessness	32 (21)
Depression	19 (12)
Suicidal ideation	18 (12)
Uncontrollable thoughts	15 (10)
Lack of agency	15 (10)
Low self-esteem	14 (9)
Isolation (loneliness)	12 (8)
Confusion	11 (7)
Self-harming	8 (5)
Experience of mental health issue	6 (4)
Mental health problem (diagnosed)	3 (2)
Total	153 (100)

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
