# Peer review of "A Look at Culture and Stigma of Suicide: Textual Analysis of Community Theatre Performances"

_ijerph, 2019, doi:10.3390/ijerph16030352_

Round 1

Reviewer 1 Report

This is a well-written manuscript describing a robust intervention that addresses a timely and pressing public health topic.  The premise and methods are strong, innovative, and well-described. The findings about the role of stigma are important and compelling. My only concerns are that the manuscript is a bit long and there are some redundancies; some minor reorganization that might improve clarity. The results section seems to include definitions of terms as well as discussion points. Sometimes this works reasonably well, but in some areas it doesn’t work that well.

I was curious to learn more about the methods used. How many people watched the performances? Where were they performed? How many performances for each play? Were actors/audience asked about their experiences doing/watching the performances? Were actors compensated for their time, either financially or by receiving academic credit? Most of the answers to these questions do not belong in this manuscript, but if they will be addressed in another paper, perhaps allude to that paper.

Specific Comments

Line 52: Is ‘reproducing’ the best word choice? I found it a bit confusing in this sentence.

91-96: Were these RQs arrived at before or after the intervention? It seems, based on subsequent discussion, that RQ3 was derived afterwards. Just be clear up front about this.

103-104. Confusing sentence as written. Is there a typo there?

232: Is it what you believed or what you hypothesized? The latter seems more appropriate.

242-3: This seems to describe recruitment; perhaps move to that section.

293-4: I found this sentence confusing. What are these comparisons, where are they presented? Is this sentence even necessary?

374-5: This concept is described in several places (see lines 400-401).

379-398: this section could be reorganized.  The ‘Here, …” on line 386 could refer to the passage before or after.  Being consistent and clear about which passage is being discussed would help.

On 395, the discussion states that it refers to a young man, but the blurb itself states parenthetically that it was a female.

414-6: this sentence is long and doesn’t seem to add much.

535-41: Defining terms here is a bit redundant and may potentially be more appropriately addressed earlier on.

631-633: this quote is redundant in its entirely.

740: typo—shouldn’t this be “locus” of control?

Author Response

MDPI Branch Office, Beijing

the IJERPH Editorial Office

Tel. +86 10-57308570; Fax: +86 10 59011089

E-mail: ijerph@mdpi.com

https://www.mdpi.com/journal/ijerph/

Jan. 8, 2019

Dear Editors,

Thank you for the constructive feedback on our qualitative textual analysis of a community-based suicide prevention program. We have explained how we have addressed each concern below. We feel the reviewers’ suggestions have greatly improved the quality of our paper and are re-submitting to the special issue on Suicide Prevention Strategies in the International Journal of Environmental Research and Public Health.

Response to Reviewer 1:

How many people watched the performances? Where were they performed? How many performances for each play? Were actors/audience asked about their experiences doing/watching the performances? Were actors compensated for their time, either financially or by receiving academic credit? Most of the answers to these questions do not belong in this manuscript, but if they will be addressed in another paper, perhaps allude to that paper.

Response: Most of these questions are addressed in another paper referenced here, Keller, Austin & McNeill, 2017. We have added language in this paper to clarify that the participants were not compensated for their time either financially or through credit. The previous paper and one new sentence in this paper (section 2.4.1) explain that writer-actors and audience members were interviewed about their experiences either performing or watching the plays, in focus group format. The plays were performed both at community theatres and high school or college campuses, and about 2-5 performances were conducted for each play. Audiences varied in size from 20-150.

Line 52: Is ‘reproducing’ the best word choice? I found it a bit confusing in this sentence.

Response: Word replaced with “amplifying.”

91-96: Were these RQs arrived at before or after the intervention? It seems, based on subsequent discussion, that RQ3 was derived afterwards. Just be clear up front about this.

Response: Yes, RQ3 was added after the intervention. The following text has been added to make this clear: “This paper addresses two research questions (RQs 1 & 2) that were developed prior to the intervention, and one research question (RQ3) that emerged afterwards.”

103-104. Confusing sentence as written. Is there a typo there?

Response: The sentence has been rewritten in the following manner: “Anthropologists have conceptualized stigma as a communal moral judgment, conferring a morally ambivalent status onto the stigmatized group [27].”

232: Is it what you believed or what you hypothesized? The latter seems more appropriate.

Response: The word “believed” has been replaced with the word “hypothesized.”

242-3: This seems to describe recruitment; perhaps move to that section.

Response: The following sentence has been deleted: ‘Student writer-actors (who wrote the scripts) had been recruited by convenience, as described above.’ And the word ‘textual’ has been added prior to the word ‘sample’ in the preceding sentence to make it clear we are talking about the content analysis only in this section.

293-4: I found this sentence confusing. What are these comparisons, where are they presented? Is this sentence even necessary?

Response: The following sentence has been deleted: ‘While results were not shared with the participants for feedback, the results were compared to focus group findings from theatre workshop participants collected during and after the project.’

374-5: This concept is described in several places (see lines 400-401).

Response: The following sentences have been deleted: ‘The interpersonal theory of suicide suggests these two factors must be present for a person to die by suicide’ and ‘These two concepts are related to stigma in that they indicate the experience of both public and internal rejection.’

379-398: this section could be reorganized.  The ‘Here, …” on line 386 could refer to the passage before or after.  Being consistent and clear about which passage is being discussed would help.

Response: The word ‘Here’ has been replaced with the phrase, ‘In the passage above.’

On 395, the discussion states that it refers to a young man, but the blurb itself states parenthetically that it was a female.

Response: The pronoun has been corrected.

414-6: this sentence is long and doesn’t seem to add much.

Response: The following sentence has been deleted: ‘This is a construct taken from Witte [61] based on her definition of the fear control response, a maladaptive response to a health threat triggered when individuals perceive their ability to control a risk as low.’

535-41: Defining terms here is a bit redundant and may potentially be more appropriately addressed earlier on.

Response: The following sentence has been added to lines 88-90: ‘This study will differentiate between views of helping others with mental health problems or depression as part of a collective responsibility, versus perspectives that view depression and alienation as a personal problem, where both the experience and the cause were focused on the individual,’ and this sentence was removed from the section on results.

631-633: this quote is redundant in its entirely.

Response: The following sentence has been removed: ‘”We are different. We don’t really open up as much as we should,” or the comment describing how getting depressed is common: “[K]ids do get depressed. It's a natural way of life -- you get depressed, you sit there and eat your carbs or whatever” (NA, high school, female).’

740: typo—shouldn’t this be “locus” of control?

Response: Thank you. Yes, the word has now been changed to “locus.”

Reviewer 2 Report

Re: ijerph-412042

This study aims to understand culture and stigma of suicide among American young adults through textual analysis of the scripts from theatre performances. Their findings addressed three research questions: 1. Stigma of mental health, including public stigma, self-stigma and stigma of help-seeking, can affect help-seeking behavior; 2. Hopelessness was mostly reported by young adults; 3. Native Americans are usually communal oriented and Caucasians are often individual oriented when talking about mental health problems. My comments are as followed.

Introduction

1.     The structure of the introduction is somewhat disorganized. The last paragraph on page 2 which talks about the research design and research questions should be moved to the end of the introduction session. On the other hand, the 2nd paragraph on page 4 which talks about the theories of cultural meaning of suicide should not be the last paragraph of the introduction session. It should be incorporated into previous paragraphs where culture is introduced.

2.     RQ2 should be put as RQ1 because it is a more general question.

3.     The authors put collective vs. individual orientation as RQ3. However, there is not any background introduction of these two concepts in the introduction session. The forming this research question is a bit unexpected.

4.     There are too many direct quotations from the reference, also in session Results and Discussion. The author may want to rephrase or paraphrase these passages.

Material and Methods

1.     The authors used theatre as their research method. However, they introduced theatre as an “intervention” (in session 1.2, 2.1 and the third paragraph in 2.4). The authors may want to provide more information to emphasize the feasibility of theatre as a research method.

2.     There are about one-third of initial participants withdrawing from the study. This is not a small proportion and the authors may want to give some explanation.

3.     In session 2.3, the authors described that researchers were “occasionally” present themselves. The definition of the frequency is not clear and how much the presentation may affect the process may need to be addressed.

4.     In the first paragraph of session 2.4, there are too many details about the researchers.

5.     Reference No.57 talks about using content analysis to study media economics. It does not support the author’s argument that a play can illuminate stigmatization of mental health and suicide. Also, this argument should be put in session 2.1 but not in 2.4 when introducing textual analysis. And again, the authors seemed to introduce a perform as an intervention which can have impact rather than as a research method. The title of reference No. 57 should be “A ‘Content’ Analysis guide…”

6.     In Line 241, sentences from “the sample consisted of…” to “recruited by convenience, as described above.” should be put in session 2.2 for introducing subjects.

7.     In session 2.4.1 and the Results session, the authors mentioned focus groups where they collected information for this study and which they compared findings with. They may want to explain more about the relationship between these focus groups and the current study. Do they have the same members? Do they have the same research purpose? Are the focus groups included in the current study?

8.     In line 273, “(n=452)” should be put right after “codable passages” in line 272 to prevent confusion.

Conclusions

1. In the first paragraph, the authors again concluded that their community-based setting as an “intervention” to provide coping strategies. It is confusing that if they want to prove that this is a way of suicide prevention or a research method of understanding suicide.

Author Response

MDPI Branch Office, Beijing

the IJERPH Editorial Office

Tel. +86 10-57308570; Fax: +86 10 59011089

E-mail: ijerph@mdpi.com

https://www.mdpi.com/journal/ijerph/

Jan. 8, 2019

Dear Editors,

Thank you for the constructive feedback on our qualitative textual analysis of a community-based suicide prevention program. We have explained how we have addressed each concern below. We feel the reviewers’ suggestions have greatly improved the quality of our paper and are re-submitting to the special issue on Suicide Prevention Strategies in the International Journal of Environmental Research and Public Health.

Response to Reviewer 2:

Introduction

1.        The structure of the introduction is somewhat disorganized. The last paragraph on page 2 which talks about the research design and research questions should be moved to the end of the introduction session.

Response: This has been done.

On the other hand, the 2nd paragraph on page 4 which talks about the theories of cultural meaning of suicide should not be the last paragraph of the introduction session. It should be incorporated into previous paragraphs where culture is introduced.

Response: The paragraph on theories of cultural meaning of suicide has been moved up into previous paragraphs on culture.

2.        RQ2 should be put as RQ1 because it is a more general question.

Response: RQs 1 & 2 have been reverse-numbered, and the corresponding discussion of results has been re-ordered.

3.        The authors put collective vs. individual orientation as RQ3. However, there is not any background introduction of these two concepts in the introduction session. The forming this research question is a bit unexpected.

Response: The following language has been added to the introduction to set up RQ3: ‘This study will differentiate between views of helping others with mental health problems or depression as part of a collective responsibility, versus perspectives that view depression and alienation as a personal problem, where both the experience and the cause were focused on the individual.’ (lines 88-91)

4.        There are too many direct quotations from the reference, also in session Results and Discussion. The author may want to rephrase or paraphrase these passages.

Response: Direct quotes have been paraphrased throughout the Introduction, Results and Discussion sections.

Material and Methods

1.        The authors used theatre as their research method. However, they introduced theatre as an “intervention” (in session 1.2, 2.1 and the third paragraph in 2.4). The authors may want to provide more information to emphasize the feasibility of theatre as a research method.

Response: Excellent point. It is clear that we have waivered between discussion of the theatre program as an intervention and research method. This ambiguity has been addressed, and the text now exclusively talks about the theatre program as a research method. The following sentence has been added to section 1.2: ‘In step with the Narrative Engagement Framework, actors in this project were instructed to use personal experiences to communicatively reduce audience members’ resistance to openly discussing the risk of suicide, the use of professional help, and the strategies needed to intervene in suicidal ideation and attempts.’

The following paragraph has been added to section 2.1: ‘Community-based programs that engage community members in storytelling and performance, such as this theatre production, offer a potentially effective approach to revealing underlying attitudes and beliefs about sensitive issues, and can be used as a research tool to generate interpersonal dialogue [35, 36]. We wanted to determine if the theatre performances were an effective method for identifying underlying barriers to and solutions for reducing the state’s suicide crisis.’

Edits have been made to the 1st and 3rd paragraphs in the Conclusion to exclusively refer to our use of theatre as a research method.

2.        There are about one-third of initial participants withdrawing from the study. This is not a small proportion and the authors may want to give some explanation.

Response: Recruitment was conducted near the end of the school year for a summer theatre program that would rely on volunteer efforts without compensation on the part of participants. Unlike traditional focus groups, the theatre research method (in our case) asked for 2-3 hours per week from participants over a 10-week period. It is not unsurprising, given the length of commitment requested, that a large portion of individuals did not complete the process. Text to address this matter has been added to 2.2.

3.        In session 2.3, the authors described that researchers were “occasionally” present themselves. The definition of the frequency is not clear and how much the presentation may affect the process may need to be addressed.

Response: Researchers were present once per week. The text has been changed accordingly. The following sentence was added to section 2.3: ‘The researchers’ presence was restricted to ensure that students had an opportunity to develop narratives without the potential inhibition caused by a researcher’s gaze.’  

4.        In the first paragraph of session 2.4, there are too many details about the researchers.

Response: This has been reduced.

5.        Reference No.57 talks about using content analysis to study media economics. It does not support the author’s argument that a play can illuminate stigmatization of mental health and suicide. Also, this argument should be put in session 2.1 but not in 2.4 when introducing textual analysis. And again, the authors seemed to introduce a perform as an intervention which can have impact rather than as a research method. The title of reference No. 57 should be “A ‘Content’ Analysis guide…”

Response: Reference No. 57 was removed. Language pertaining to the project as an intervention has been eliminated. The rationale for using textual analysis was moved to 2.1.

6.        In Line 241, sentences from “the sample consisted of…” to “recruited by convenience, as described above.” should be put in session 2.2 for introducing subjects.

Response: Language about participants has been moved to 2.2, and the sentence in 2.4 has been edited to clarify that we are referring to the sample of text: ‘The textual sample consisted of the entire population of such plays produced by the project under study.’

7.        In session 2.4.1 and the Results session, the authors mentioned focus groups where they collected information for this study and which they compared findings with. They may want to explain more about the relationship between these focus groups and the current study. Do they have the same members? Do they have the same research purpose? Are the focus groups included in the current study?

Response: The focus groups were part of a separate study referenced here and reported fully in Keller, Austin & McNeill, 2017. Language has been added to section 2.4.1 to clarify the relationship between the two studies: ‘The focus groups were part of a separate study examining the impact of the plays on audience members and writer-actors, addressing research questions pertaining to whether the plays increased perceived self-efficacy to seek help. While some of the participants were the same as those used in this study, this study focused exclusively on comments articulated through the playwriting process, while the focus group study analyzed comments collected after the plays were completed.’

8.     In line 273, “(n=452)” should be put right after “codable passages” in line 272 to prevent confusion.

Response: This was done.

Conclusions

1.      In the first paragraph, the authors again concluded that their community-based setting as an “intervention” to provide coping strategies. It is confusing that if they want to prove that this is a way of suicide prevention or a research method of understanding suicide.

Response: Language pertaining to the theatre project as an intervention has been removed.

Dr. Sarah N. Keller, M.S., Ph.D.

Department of Communication & Theatre

Montana State University Billings

Billings, MT 59101

(406) 896-5824

(406) 657-2178, Administrative Assistant

skeller@msubillings.edu

Round 2

Reviewer 2 Report

Re: ijerph-412042- Revised Version Review Request

The authors have addressed most of my concerns.

However, I still have the last concern. It is great that the authors have made a clarification that they used theatre program as a research method rather than an intervention. But in their revised paragraph in section 2.1, they added that “We wanted to determine if the theatre performances were an effective method…” and “community theatre of this type can provide a research tool for …” in conclusions. I would suggest that the authors may think over their research aims. In the introduction, they have set up three research questions that the research aimed to answer. Proving that theatre is an effective method was not included in their research questions/aims. Do the authors want to include proof of the effectiveness of theatre as a research method in their research goals? Thus, according language and presentation related to this aim should be added throughout the manuscript. Or do the authors simply use theatre as an approved research method to assess mental health issues? Then, they should provide enough evidence/literature to justify their usage of theatre as an appropriate research method.

Author Response

This reviewer has pointed out an ambivalent portion of our study description, and suggested an easy fix. It is true that we have simply used theatre as an approved research method and are not setting out to prove its effectiveness. We have tweaked the language in sections 1.2 and 2.1 accordingly, and added literature to support this approach. In section 1.2, we have re-titled the section to be:

“1.2. Approaches to Shed Light on Stigma” (rather than “Interventions to Decrease Stigma”) and revised the last sentence of 1.2 to say:

[actors were instructed to] communicatively engage audience members around the topics of…

We have added the following studies to support the use of theatre as a research method:

Conrad, D. (2004). Exploring risky youth experiences: Popular theatre as a participatory, performative research method. International Journal of Qualitative Methods3(1), 12-25.

Webster, L., & Mertova, P. (2007). Using narrative inquiry as a research method: An introduction to using critical event narrative analysis in research on learning and teaching. Routledge.

Additionally, we have edited the last 2 sentences of section 2.1 to report:

Our team believed (rather than hypothesized) that a play… would illuminate community-specific factors associated with stigmatization of suicidal conditions.

We used the theatre performance as a method for identifying underlying barriers to and solutions for reducing the state’s suicide crisis.

The above re-phrasing is intended to clarify that we are simply using theatre as a research method, rather than attempting to prove its effectiveness, which we believe is beyond the scope of this study. Thank you for helping to clarify our methodology and make this paper stronger.